# Gender differences in clinical presentation and illicit substance use during first episode psychosis: a natural language processing, electronic case register study

Jessica Irving  ,[1] Craig Colling,[2] Hitesh Shetty,[2] Megan Pritchard,[2,3] Robert Stewart,[2,3] Paolo Fusar-Poli,[2] Philip McGuire,[1,2] Rashmi Patel  [1,2]

¹Department of Psychosis Studies, Institute of Psychiatry Psychology and Neuroscience, London, UK
²Biomedical Research Centre, South London and Maudsley NHS Foundation Trust, London, UK
³Department of Psychological Medicine, Institute of Psychiatry Psychology and Neuroscience, London, UK

**Correspondence to**
Jessica Irving;
jessica.irving@kcl.ac.uk

## ABSTRACT

**Objective** To determine whether gender differences in symptom presentation at first episode psychosis (FEP) remain even when controlling for substance use, age and ethnicity, using natural language processing applied to electronic health records (EHRs).

**Design, setting and participants** Data were extracted from EHRs of 3350 people (62% male patients) who had presented to the South London and Maudsley NHS Trust with a FEP between 1 April 2007 and 31 March 2017. Logistic regression was used to examine gender differences in the presentation of positive, negative, depressive, mania and disorganisation symptoms.

**Exposure(s) (for observational studies)** Gender (male vs female).

**Main outcome(s) and measure(s)** Presence of positive, negative, depressive, mania and disorganisation symptoms at initial clinical presentation.

**Results** Eight symptoms were significantly more prevalent in men (poverty of thought, negative symptoms, social withdrawal, poverty of speech, aggression, grandiosity, paranoia and agitation). Conversely, tearfulness, low energy, reduced appetite, low mood, pressured speech, mood instability, flight of ideas, guilt, mutism, insomnia, poor concentration, tangentiality and elation were more prevalent in women than men. Negative symptoms were more common among men (OR 1.85, 95% CI 1.33 to 2.62) and depressive and manic symptoms more common among women (OR 0.30, 95% CI 0.26 to 0.35). After adjustment for illicit substance use, the strength of associations between gender and negative, manic and depression symptoms increased, whereas gender differences in aggression, agitation, paranoia and grandiosity became insignificant.

**Conclusions** There are clear gender differences in the clinical presentation of FEP. Our findings suggest that gender can have a substantial influence on the nature of clinical presentation in people with psychosis, and that this is only partly explained by exposure to illicit substance use.

## Strengths and limitations of this study

► Our methodology using natural language processing (NLP) on an electronic case register allows us to obtain the largest sample size to date in this field.

► This is the first study to control for onset age, ethnicity and substance use in gender differences in symptom presentation in first episode psychosis.

► By examining patients at first presentation, our findings represent differences early on in the trajectory of psychosis that are less likely to be affected by treatment.

► NLP techniques are associated with a degree of measurement error, which could impact the recorded frequencies for each symptom.

► Symptoms reported in electronic health records might be impacted by gender biases among clinicians.

## INTRODUCTION

The incidence and clinical presentation of first episode psychosis (FEP) varies by gender.[1] Some of these differences, such as earlier age of onset for men, and a bimodal age distribution for women, are well established.[2–4] Others, such as symptoms at FEP presentation, are less well established.

Previous studies report that male patients with FEP present with greater negative and disorganisation symptom burden and greater illicit substance use than female patients.[5] Female patients conversely appear to present with more affective symptoms and parasuicidal behaviour than male patients, with affective symptoms increasing risk for parasuicidal behaviour.[6 7] Despite these findings, several studies have failed to find significant clinical differences in symptoms at all.[8 9]

At population level, men show greater risk for substance use than women,[10] and this pattern is especially prominent in psychosis samples.[11–13] Negative symptoms, such as motivational deficits, are common to both

psychosis and substance use disorders[14]; increased prevalence of negative symptoms in men could therefore plausibly be associated with illicit substance misuse.[3] However, no published studies have attempted to control for illicit substance use at the point of an emerging psychotic disorder.

Few studies investigating variation in psychotic symptoms at first presentation control for ethnicity. Both psychosis risk and symptomatology show ethnic variability[1 15–17]; furthermore, specific genders and ethnic groups are associated with unique and shared risk factors for psychosis.[18] The way in which these factors might interact in psychosis is unknown.[19]

Quantifying the impact of gender on symptom expression can provide a clue as to the underlying mechanisms. For example, negative symptoms are difficult to treat and correlate with poorer outcomes.[20] The cause of negative symptoms remains unknown, and this is the rate-limiting obstacle to developing new treatments. If these are genuinely more common in men, even factoring in increased substance use in men, it suggests that gender-related biological factors may play a role in their pathophysiology.[3] For example, sex-specific hypothalamic–pituitary–gonadal dysfunction are implicated in psychosis pathophysiology[21]; oestrogens are known to modulate the dopaminergic and glutamatergic systems, both of which show aberrant functioning in psychosis.[22] More recently, central nervous system autoimmunity has been implicated in the aetiology of psychosis. Around 80% of patients with anti-$N$-methyl-$D$-aspartate receptor encephalitis are women whereas anti-LGI1 and anti-CASPR2 encephalitis more frequently occur in men.[23] Understanding these gender differences might inform the development of novel, more targeted treatments for psychosis.

We sought to investigate gender differences in psychosis symptoms experienced by those presenting with FEP to a specialised early intervention service (EIS) using natural language processing (NLP) techniques applied to electronic health record (EHR) data. We hypothesised that there would be differences between the genders in terms of symptoms and substance use recorded, and that these might be impacted by age at onset, ethnicity and substance use. NLP techniques applied to free-text EHR data can automatically classify patients who present with specific symptoms and other characteristics.[24] These techniques use machine learning to learn from human-annotated examples and are able to navigate term negation (eg, 'No thought disorder elicited') and mentions irrelevant to that which is being measured (eg, 'His father has thought disorder'). This approach has previously been used to investigate transdiagnostic risk for psychosis,[25] the association of negative symptoms with antipsychotic treatment failure,[26] poor insight as a predictor of service use outcomes[27] and the association of cannabis use with hospital admissions[28] in people with psychotic disorders. Our sample of 3350 patients represents the largest known sample in this field to date.

## METHODS

### Reporting
We used the STROBE cross-sectional checklist when writing our report.[29]

### Study setting
Clinical data were obtained from de-identified EHRs held by the South London and Maudsley (SLaM) NHS Trust using the Clinical Record Interactive Search (CRIS) system. SLaM is one of the largest mental healthcare providers in Europe and implemented a fully EHR system from 2006 onwards. The Trust holds records for over 450 000 patients. Its provision includes four early intervention services in its catchment area of four boroughs (Croydon, Lambeth, Lewisham and Southwark) in southeast London with a population of around 1.2 million residents.

As EHRs provided limited structured information on symptomatology and substance use, a suite of NLP tools has recently been developed within CRIS to allow conversion of unstructured free text, such as that of uploaded attachments and discharge summaries, into structured data. These yield complementary information on diagnoses, symptomatology and other patient characteristics.

### Sample
Data were extracted for all individuals aged 16–65 years accepted to EISs for FEP within the SLaM catchment area (Lambeth Early Onset team, Southwark Team for Early Psychosis, Lewisham Early Intervention Service and Croydon Outreach and Assertive Support Team) between 1 April 2007 and 31 March 2017.[30] From 2016, the upper age limit of EIS increased from 35 to 65. The original sample contained 3597 individuals. Individuals with no symptom data (n=63) and missing ethnicity data (n=70) were excluded, leaving a sample size of 3464 participants with complete covariate data and at least one documented symptom. Inspection of acceptance date distribution revealed mass uploads of backlogged records on three outlier days in April 2007 for one service; we therefore removed these individuals (n=114) from analysis as they were not genuine first episodes.

### Measures

#### Demographics
Structured data were extracted on age at accepted referral, gender and ethnicity. Ethnicity was recoded as Asian, Black—African, Black—Caribbean, Black—Other, Mixed, other and White (online supplemental table 1).

#### Symptoms and substance use
CRIS offers a suite of NLP algorithms that are able to create structured data from unstructured free text through detection of key words and phrases. In brief, the algorithms are developed by applying cross-validated support vector machines to a human-annotated training corpus of 'positive' (eg, 'urinary drug screen shows cocaine'), 'negative' (eg, 'does not use cocaine') and 'irrelevant' (eg, 'cocaine is a highly addictive drug') examples for each symptom and substance. Their performance is then

quantified against unseen data in an iterative process of development and testing.[24] The full library of NLP algorithms available within CRIS and details of their development are provided on the CRIS website.[31]

These algorithms were used to extract data for 42 symptom constructs and 4 substance use indicators mentioned in the period 3 months either side of the date of each patient's acceptance to an EIS clinical team. Dichotomous variables were created indicating the presence or absence of at least one mention of a given symptom or substance use indicator. We excluded symptoms of low prevalence (less than 0.05% of the sample). Symptoms were classified into the following domains identified in a principal axis factor analysis by Demjaha and colleagues[32] using the approach employed by Jackson et al[24] (see online supplemental table 2): (1) positive, (2) negative, (3) disorganisation, (4) manic and (5) depressive symptoms. *Insight* was assigned to (6) other. Extracted substance data concerned cocaine, cannabis, amphetamine and methylenedioxymethamphetamine (MDMA).

NLP algorithm performance is mainly measured using precision (proportion of relevant instances among retrieved instances) and recall (proportion of relevant instances retrieved from all relevant instances in data). Precision estimates were generated through manual validation of each symptom (0.64–0.99, mean=0.86) and substance algorithm (0.87–0.97, mean=0.92). NLP algorithms are supplied in online supplemental table 3.[24]

### Diagnosis

We recorded each patient's International Classification of Diseases (ICD-10) diagnosis closest to 12 months from the EIS accepted date. Patient diagnoses were derived from either a structured primary diagnosis field or unstructured free text using NLP. Psychosis diagnoses were recoded as bipolar disorder, drug-induced psychosis, schizophrenia, psychotic depression, schizoaffective disorder or other psychosis (online supplemental table 4).

### Statistical analysis

All analyses were conducted using R V.3.6.0. We obtained descriptive statistics for demographical characteristics, diagnosis at 12 months and symptom and substance use distributions.

Binary logistic regression was used to assess gender differences in symptomatology and substance use. Subsequent multivariable models controlling for age and ethnicity were constructed to examine potential confounding effects in differences in symptoms or substance use. The reference group for ethnicity was 'White', which comprised the sample majority. P values were adjusted for multiple testing using the Benjamini-Hochberg correction (false discovery rate (FDR))[33] using the *p.adjust* package in R.

The adjusted models were then updated to include illicit substance use as an additional covariate. A dichotomous variable was created to flag any mention of cannabis, cocaine, MDMA or amphetamine as *present/absent*.

**Table 1** Sample characteristics

| Characteristic | n | Male | Female |
|---|---|---|---|
| N (%) | 3350 | 2092 (62) | 1258 (38) |
| Age at referral (median, IQR) | 24 (20–29) | 23 (20–28) | 25 (21–30) |
| Ethnicity (N, %) | | | |
| Asian | 234 | 144 (7) | 90 (7) |
| Black—African | 635 | 383 (30) | 252 (20) |
| Black—Caribbean | 216 | 123 (6) | 93 (7) |
| Black—other | 847 | 533 (3) | 314 (25) |
| White | 1037 | 652 (49) | 385 (31) |
| Other | 381 | 257 (12) | 124 (10) |
| Early intervention team | | | |
| Lambeth | 1084 | 683 (33) | 401 (32) |
| Southwark | 826 | 520 (25) | 306 (24) |
| Lewisham | 678 | 400 (19) | 278 (22) |
| Croydon | 761 | 495 (24) | 266 (21) |

Given that p values for male gender increased when controlling for substance use, we ran post hoc analyses in which we (1) ran diagnostic tests for multicollinearity by calculating the variance inflation factor (VIF) for each predictor in each model and (2) built an additional model inclusive of a gender*substance use interaction term.

Multinomial logistic regression, implemented via the *nnet* package, was used to investigate gender differences in diagnosis at 12 months from the patient's first accepted referral to an EIS. We controlled for age and ethnicity.

### Patient and public involvement

Project applications requiring access to CRIS data are subject to approval from the CRIS Oversight committee, which is service user led.

### RESULTS

Table 1 presents sample characteristics. The final sample comprised 3350 patients of predominantly male gender (n=2077, 62%). Median (IQR) age was 24 (20–29). White (n=1037), black—other (n=847) and black—Caribbean (n=635) were the most frequently represented ethnic groups.

Age distribution by gender and by substance use are presented in figure 1. Number of referrals sharply inclines in the late teenage years for both genders. Both genders experience a sharp peak in incidence in the early twenties which falls at a greater rate for male patients than female patients. Individual density plots showing referral numbers by age stratified by amphetamine, cannabis, cocaine and MDMA use are provided in online supplemental figure 1. The age by cannabis use plot most closely mirrors the distribution of overall substance use shown below. Referrals for cannabis and amphetamine users show inversed patterns around age 30.

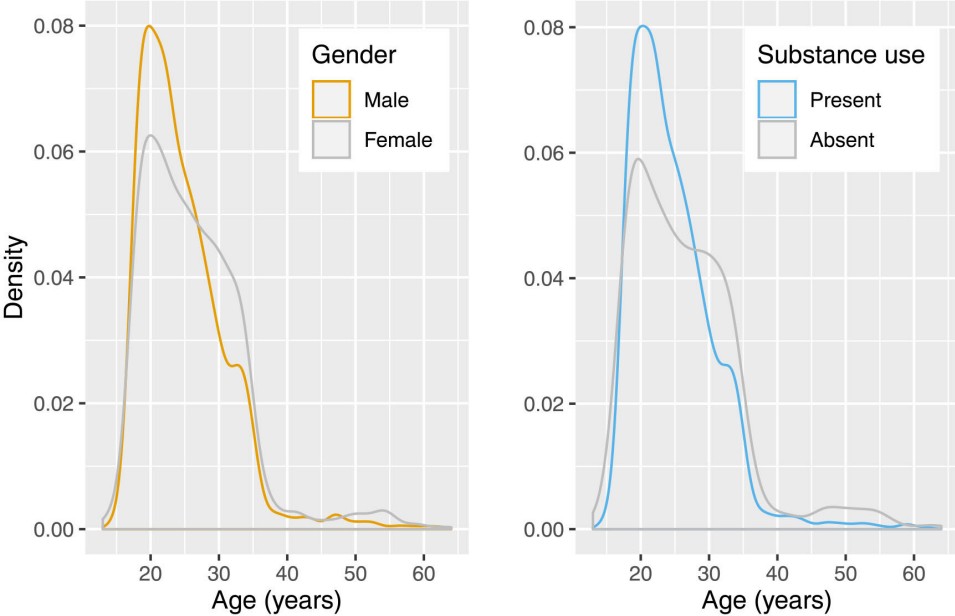

**Figure 1** These density plots present age at admission by gender (left) and substance use (right). Density plots—a variation of the histogram—present the distribution of numerical data but with kernel smoothing to reduce the effect of noise in the data, thereby creating a smoother line. The peaks of a density plot indicate where values are concentrated.

Figure 2 presents gender differences in symptomatology and substance use obtained via binary logistic regressions before and after adjustment for onset age, ethnicity and illicit substance use. ORs refer to male gender as the reference group. All values and overall ORs for each symptom group are provided in online supplemental tables 5 and 6.

Of 42 symptoms investigated, 8 were more prevalent in men and 14 more prevalent in women before adjustment for age, ethnicity and substance use. Highly significant gender differences (p≤0.001) existed for poor appetite, low energy, tearfulness, guilt, mutism, grandiosity, negative symptoms, social withdrawal, aggression, mood instability and pressured speech. Symptoms more strongly associated with male gender included (in order of decreasing strength of association) poverty of thought, negative symptoms (general), social withdrawal, poverty of speech, aggression, grandiosity, paranoia and agitation. Male gender was also significantly associated with cannabis, cocaine and amphetamine use.

Female patients were significantly more likely to have depressive and manic symptoms recorded than male patients. In order of decreasing strength, female gender was associated with tearfulness, low energy, reduced appetite, low mood, pressured speech, mood instability, guilt, mutism, insomnia, poor concentration and elation.

Delusions and auditory and visual hallucinations did not differ significantly between genders (p>0.05), but olfactory, tactile and gustatory hallucinations were reported more frequently in female patients. Flight of ideas and tangential speech were more frequently recorded in women than men, but no other disorganisation symptoms showed significant gender differences.

All significant differences (bar tangential speech) remained after adjustment for age and ethnicity (see online supplemental table 4). After further adjustment for illicit substance use, gender differences in aggression, agitation, paranoia and grandiosity became non-significant (p>0.05). Significant differences in negative and depression symptoms remained; these symptoms showed increased log odds. Some previously insignificant gender differences (poor insight, disturbed sleep and irritability) became significantly more likely in women. Elation, flight of ideas and tangential speech increased in significance. Gender differences were driven by individual, rather than broad groups of symptoms (with the exception of disorganisation; online supplemental table 5). No model variables showed VIF values above 1.08, indicating no issues with multicollinearity (online supplemental table 6). Interaction effects are presented in online supplemental table 5.

There were clear gender differences in diagnosis 12 months from service accepted date, which were robust when adjusting for age and ethnicity (online supplemental table 7). Men were three times more likely to receive a non-schizophrenia, updated diagnosis of drug-induced psychosis than women, and more than half as likely to receive a diagnosis of bipolar disorder than women.

## DISCUSSION

The present study investigated gender differences in symptomatology for 3350 patients presenting with FEP. We found clear gender differences in reported symptoms, of which almost all remained when controlling for age

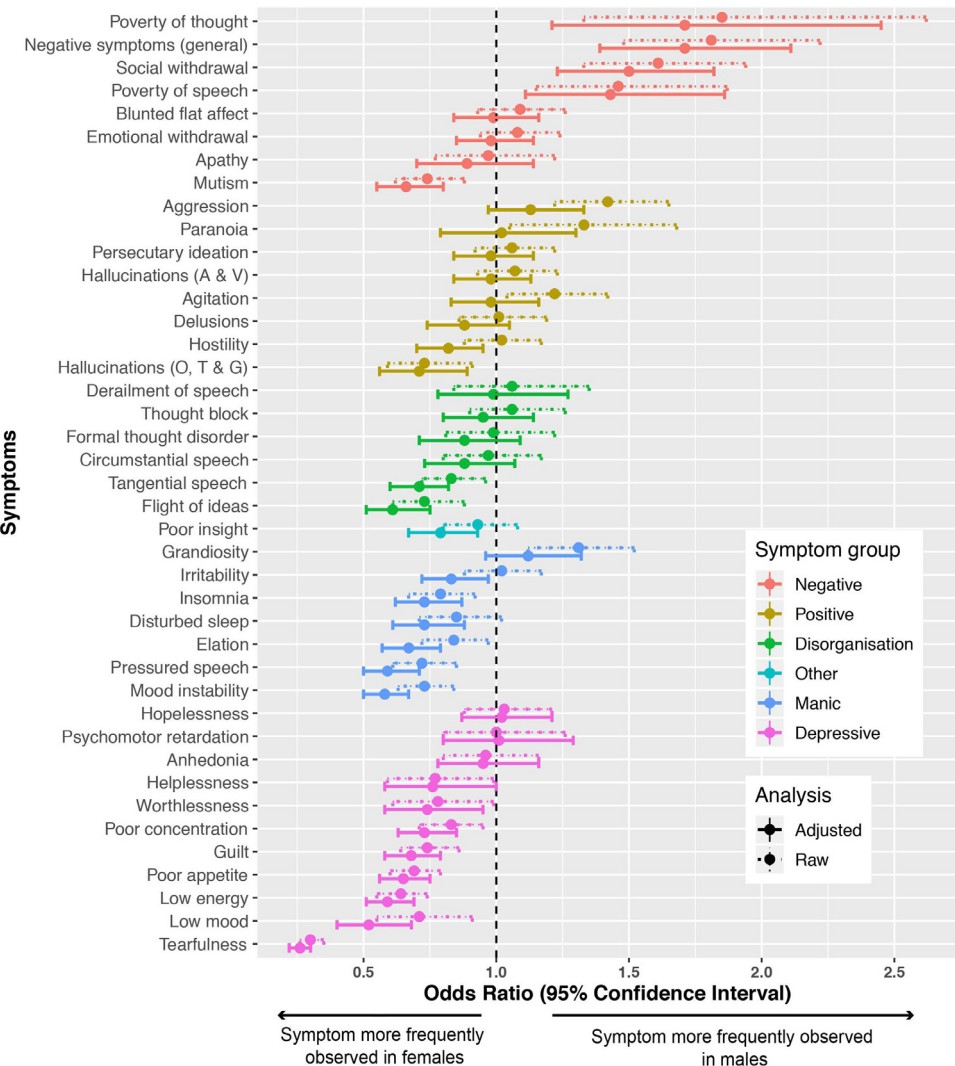

**Figure 2** Odds ratios (ORs) greater than 1 indicate that a symptom is more likely to be observed in men than women, and vice versa for ORs less than 1. Dashed lines indicate ORs and CIs before adjustment for covariates, with solid lines indicating estimates after adjustment. Hallucinations (A & V) refer to auditory and visual hallucinations; hallucinations (O, T & G) refer to olfactory, tactile and gustatory hallucinations; ORs, Odds ratios.

and ethnicity. However, controlling for illicit substance use resulted in several changes to associations of gender with reported symptoms.

In univariate analyses, most negative symptoms, and use of all substances, were reported more frequently for male patients. Female patients showed more manic (bar grandiosity) and depressive symptoms. Our results are consistent with studies that find more negative symptoms and substance use in male FEP patients, and more affective symptoms in female patients.[5 6 12] Many of these differences are found in the general population, such as increased affective symptoms in women, and substance abuse in men.[10 34] The findings for disorganisation symptoms were less clear. Where these existed, they tended towards female patients, in contrast to the findings of Thorup and colleagues.[5]

Strikingly, all significant gender differences (bar tangential speech) remained after controlling for age and ethnicity. This suggests that gender differences in

symptoms at presentation are independent of age and ethnicity. To the best of our knowledge, no studies have accounted for ethnicity in analyses of gender symptom differences,[5 11 12 35] despite previous research indicating a significant association between ethnicity and risk of psychosis.[36] The lack of effect of referral age on gender differences is consistent with other studies' findings.[6 12 35]

After adjusting for illicit substance use, age and ethnicity, negative symptoms were still more prominent in men; manic and depressive symptoms became even more prominent in women. This suggests that gender differences in these three symptom domains reflect genuine sex differences in pathophysiology. Biologically, endocrine variation (particularly hypothalamic–pituitary–gonadal dysfunction in both sexes) has been implicated in gender differences of psychosis presentation. For example, converging epidemiological, clinical and animal research lends support to a neuroprotective effect of estrogens in women as a buffer against illness

development and severity.[37] Interestingly, symptom severity appears to increase in the low oestrogen phase of the menstrual cycle and during the menopause. Oestrogen's neuroprotective effects have been attributed to its effects on the dopaminergic and glutamatergic systems, which are both implicated in psychosis.[21] Anatomically, disrupted sexual dimorphism of several regions in the brain has been repeatedly observed in psychosis.[3] Men and women are also differentially exposed to psychosocial risk factors for psychosis (recently summarised in an umbrella review).[1] The immune system also varies by gender, and autoimmune central nervous system disorders that show associations with psychosis vary in incidence between men and women.[23]

While referral numbers in cannabis users peak at age 21, those in amphetamine users have bimodal peaks at ages 22 and 29. As our sample already have psychosis at first assessment, any inferences drawn from these findings are speculative. It is possible that patients with cannabis use are presenting with psychosis at a younger age, which is consistently replicated in the literature.[38] This finding could also reflect differential age trends for usage of amphetamines and cannabis, with amphetamine more popularly used at a later age than cannabis. Conversely, both drugs might be used at comparable rates by both age groups, with amphetamine use demonstrating a larger effect in older age groups. It could also be that cannabis use confers the biggest immediate risk to the adolescent brain whereas amphetamine use has longer term effects.

Our findings suggest that substance use plays an important role in clinical presentation of some symptoms, with differential effects between genders. Gender differences in aggression, agitation, paranoia and grandiosity all became insignificant, which might be explained by their associations with illicit substance use. This suggests that clinicians should take the role of illicit substance use into account when identifying strategies for symptomatic relief. In general, there is little research available into the ways in which substance use affect FEP presentation at the symptom level. Consistent with previous findings, male FEP patients were far more likely to use cannabis, cocaine and amphetamine than female FEP patients.[34] Comorbid substance use in these populations is associated with medication non-compliance, greater risk of relapse, higher suicidality and poorer outcomes in symptoms and functioning overall.[39–41]

Introduction of illicit substance use as a covariate caused the pattern of observed gender differences to change considerably, often inflating significance. Model VIF values close to one discount multicollinearity as a plausible mechanism for inflated probability values. One Danish study has provided initial evidence of a role for substance use in gender differences in clinical presentation in 578 patients assessed 5 years on from first episode.[5] The authors found significant associations between both male gender and substance abuse for negative and disorganisation symptoms; controlling for substance use reduced effect sizes for male gender and removed prior

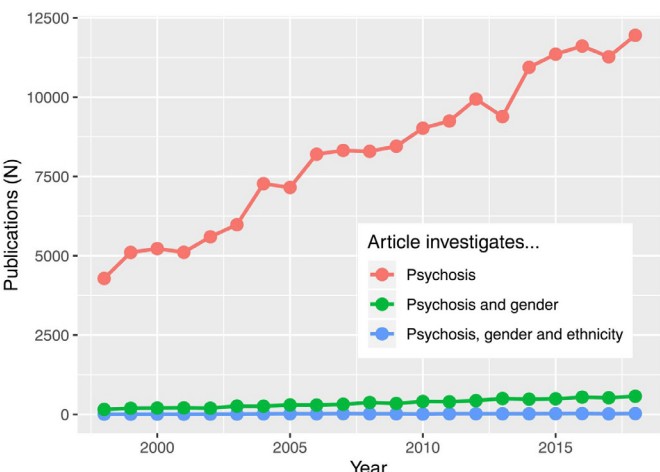

**Figure 3** As interest in psychosis has grown over time, few studies have examined the role of gender, and scarcely any have examined gender and ethnicity together. Source: Web of science (accessed January 2020).

effects of male gender on global assessment of functioning symptom score.

In general, the impact of gender on clinical presentation and outcomes in FEP is under-researched. While 45 944 papers were published on psychotic disorders between 1995 and 2018, only 291 (0.01%) specifically investigated gender differences (figure 3).

This reflects other areas of medical research where the importance of gender differences are increasingly recognised, such as cardiovascular disease,[42] where clinical presentation and symptoms vary by gender leading to underdiagnosis of heart disease and increased mortality in female patients.[43] Our findings highlight the need to investigate the impact of gender differences on clinical presentation and their underlying pathophysiology, which could affect treatment outcomes following the onset of FEP.

### Strengths and limitations

There are some limitations to our approach. Symptoms of NLP algorithms vary in precision and recall performance; this variability in NLP accuracy could impact the recorded frequencies of each symptom.[24 28] However, one would expect measurement error to be spread equally across genders. Furthermore, the algorithms cannot be used to infer symptom severity, as patients who are more unwell or represent complex cases typically have more documentation. Therefore, gender differences in symptoms may not generalise across the range of disease severity. There was no significant difference in document count between genders, therefore this is unlikely to bias results. We had no healthy control group and therefore cannot establish whether gendered symptom differences are specific to FEP or extend to the general population. Our methodology is also unable to identify the extent to which the gender differences observed represent sex differences at the biological level. Given service changes in 2016, there is only 1 year of data available for individuals aged 36 and

older. Finally, symptoms reported in EHRs might to some extent reflect gender biases in recording or patients' willingness to disclose.

Despite these limitations, our approach also has its strengths. To the best of our knowledge, this is the largest study to investigate gender differences in FEP, and the only study to control for ethnicity and substance use in doing so. By examining patients at first presentation, our findings represent differences early on in the trajectory of psychosis that are less likely to be affected by treatment.

## Conclusion

We found that the clinical presentation of FEP varies by gender and that this is only partly explained by exposure to illicit substances. Most research on gender differences in psychosis has focused on candidate pathways to psychosis as a diagnostic construct. Studies are now urgently needed to link biological and psychosocial risk factors to specific symptoms so that novel, more targeted treatments can be developed.

**Contributors** The study was conceived by RP and JI. Data extraction and statistical analysis were performed by JI with support from HS, MP and RP. Reporting of findings was carried out by JI and RP. All authors (JI, CC, HS, MP, RS, PF-P, PM and RP) contributed to study design, manuscript preparation and approved the final version.

**Funding** HS, MP, RS and PM receive funding from the National Institute for Health Research (NIHR) Biomedical Research Centre at South London and Maudsley NHS Foundation Trust and King's College London, which also supports the development and maintenance of the BRC Case Register. RP has received support from a Medical Research Council (MRC) Health Data Research UK Fellowship (MR/S003118/1) and a Starter Grant for Clinical Lecturers (SGL015/1020) supported by the Academy of Medical Sciences, The Wellcome Trust, MRC, British Heart Foundation, Arthritis Research UK, the Royal College of Physicians and Diabetes UK.

**Competing interests** All authors have completed the ICMJE uniform disclosure form at www.icmje.org/coi_disclosure.pdf and declare: RS has received research funding from Roche, Janssen, GSK and Takeda. PFP has received grant funds from Lundbeck and honoraria from Lundbeck, Menarini and Angelini outside the current study. RP has received grant funds from Janssen and consultancy fees from Induction Healthcare and Holmusk. The other authors declare no competing interests.

**Patient consent for publication** Not required.

**Ethics approval** The SLaM Biomedical Research Centre Case Register and CRIS have received ethical approval from the Oxfordshire Research Ethics Committee C (18/SC/0372) as an anonymised dataset for mental health research. As CRIS data are de-identified, individual consent was not sought.

**Provenance and peer review** Not commissioned; externally peer reviewed.

**Data availability statement** Data may be obtained from a third party and are not publicly available. Data sharing: The data accessed by CRIS remain within an NHS firewall and governance is provided by a patient-led oversight committee. Subject to these conditions, data access is encouraged and those interested should contact RS (robert.stewart@kcl.ac.uk), CRIS academic lead.

**ORCID iDs**
Jessica Irving http://orcid.org/0000-0002-2847-6508
Rashmi Patel http://orcid.org/0000-0002-9259-8788

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
