## [Reviewer comments · BMJ Open]

ARTICLE DETAILS

TITLE (PROVISIONAL)	Gender differences in clinical presentation and illicit substance use during first episode psychosis: a natural language processing, electronic case register study
AUTHORS	Irving, Jessica; Colling, Craig; Shetty, Hitesh; Pritchard, Megan; Stewart, Robert; Fusar-Poli, Paolo; McGuire, Philip; Patel, Rashmi

VERSION 1 – REVIEW

REVIEWER	muralidharan kesavan National Institute of Mental Health and Neuro Sciences, Bangalore, India.
REVIEW RETURNED	15-Sep-2020

GENERAL COMMENTS	the authors have presented a secondary analysis of existing data from the health records of persons presenting with first episode psychosis to understand the role of gender in the differential symptom presentation of FEP the findings are interesting the statistical analysis and methodology is reasonably well described the strengths and limitations are clearly described the neurobiological explanation that has been attempted in the discussion section appears too hypothetical and theorised; which is not in keeping with the objective of this analysis. that can be done away with. the paper reads well overall.
--

REVIEWER	Christian Núñez Institute of Biomedical Research Sant Pau, Barcelona, Spain
REVIEW RETURNED	31-Jan-2021

GENERAL COMMENTS	Irving et al. investigated differences in symptom presentation between male and female patients with first episode psychosis. They identified several symptoms to have a different prevalence in men and women, as well as sex differences in diagnosis after 12 months. Importantly, most of these differences were still significant after controlling for age, substance use, and ethnicity. They employed a natural language processing (NLP) algorithm to read electronic health records, which allowed them to attain a very large sample of patients. Overall, the paper is well written and presented. It will surely be a nice addition to the current literature on this topic. It is remarkable that they controlled their results for substance use and ethnicity, which are two very important factors that most studies on the field tend to omit.
--

	I have only a few comments/suggestions of minor relevance:  - I suggest replacing the word “gender” by “sex” throughout the manuscript. “Gender” is rather a social construct, while “sex” is more of a biological term (see, for example, this link: https://cihr-irsc.gc.ca/e/48642.html). Unless you had specific information on whether the patients identified themselves as men or women, using the word “sex” would be more accurate. - What was the sample size included in the final analyses? In the Abstract and at the beginning of the Results and Discussion sections, it is stated that it was 3,340 but, according to the information provided in the Methods and Table 1, I guess the correct number is 3,350. - It would be useful to provide a little bit more thorough description of the NLP algorithm employed. I am not an expert in NLP, and I wonder how the algorithm deals with those cases in which a particular word appears in the middle of a negative statement (e.g., “the patient reported no cannabis consumption”) in the electronic record. I assume the algorithm is able to detect and do a correct classification of such instances but adding more information about how the algorithm works may help the readers. Alternatively, a link to the algorithm, if it is open-source, or a reference to a previous manuscript in which this algorithm is described in depth, could be useful as well.
--	--

REVIEWER	Carla Comacchio Aulss 9 Verona, Italy
REVIEW RETURNED	02-Feb-2021

GENERAL COMMENTS	This is a study on the impact of gender and substance abuse on symptomatology at psychosis onset. I have the following observations: INTRODUCTION:  -NLP techniques are just mentioned, but a short paragraph on them could be useful -objectives are not clearly stated and hypotheses are completely missing METHODS:  -specify here which substances are detected by your algorithms (they are specified somewhere else in the paper) DISCUSSION:  -please, expand the section on ethnicity and age of psychosis onset, especially if they are included in your objectives -the section on gender differences in symptoms presentation that become insignificant after controlling for substance abuse is nice and could be discussed more extensively
--

VERSION 1 – AUTHOR RESPONSE

Reviewer: 1

Dr. Kesavan Muralidharan, National Institute of Mental Health and Neurosciences, Karnataka, India

Reviewer: 1

Competing interests of Reviewer: None declared

Comments to the Author:

the authors have presented a secondary analysis of existing data from the health records of persons presenting with first episode psychosis to understand the role of gender in the differential symptom presentation of FEP

the findings are interesting

the statistical analysis and methodology is reasonably well described

the strengths and limitations are clearly described

the neurobiological explanation that has been attempted in the discussion section appears too hypothetical and theorised; which is not in keeping with the objective of this analysis. that can be done away with.

the paper reads well overall.

/* Thank you for your supportive comments. We present neurobiological research to provide context to our finding that some of our identified gender differences in negative, manic and depressive symptoms may represent biological sex differences in pathophysiology, and to establish that there is an existing evidence base of candidate mechanisms for researchers interested in taking our work further. The contextual literature we present here builds on that identified in the Introduction. However, in light of your comments we have added a sentence to the Limitations to highlight that our methodology does not enable us to make conclusive inferences regarding biological differences: "Our methodology is also unable to identify the extent to which the gender differences observed represent sex differences at the biological level." (Page 14) */

Reviewer: 2

Dr. Christian Núñez, Hospital de la Santa Creu i Sant Pau Institut de Recerca

Comments to the Author:

Irving et al. investigated differences in symptom presentation between male and female patients with first episode psychosis. They identified several symptoms to have a different prevalence in men and women, as well as sex differences in diagnosis after 12 months. Importantly, most of these differences were still significant after controlling for age, substance use, and ethnicity. They employed a natural language processing (NLP) algorithm to read electronic health records, which allowed them to attain a very large sample of patients.

Overall, the paper is well written and presented. It will surely be a nice addition to the current literature on this topic. It is remarkable that they controlled their results for substance use and ethnicity, which are two very important factors that most studies on the field tend to omit.

I have only a few comments/suggestions of minor relevance:

- I suggest replacing the word "gender" by "sex" throughout the manuscript. "Gender" is rather a social construct, while "sex" is more of a biological term (see, for example, this link:

<https://eur03.safelinks.protection.outlook.com/?url=https%3A%2F%2Fcihr-irsc.gc.ca%2F%2F48642.html&data=04%7C01%7Cjessica.irving%40kcl.ac.uk%7C5f1c2e2893>

a6448349c508d8d1997768%7C8370cf1416f34c16b83c724071654356%7C0%7C0%7C637489804352173321%7CUnknown%7CTWFpbGZsb3d8eyJWljiMC4wLjAwMDAiLCJQIjoiV2luMzliLCJBTiI6Ij1haWwiLCJXVCi6Mn0%3D%7C1000&data=gOHW033e3uhyi%2Bvd7EAzpEXzLuo9sr5Yq8fDVtm8M1Y%3D&reserved=0). Unless you had specific information on whether the patients identified themselves as men or women, using the word “sex” would be more accurate.

/* Thank you for your supportive comments regarding our methodology. We used the term “Gender” as this is the term employed in the electronic health record as defined in the NHS data dictionary (https://datadictionary.nhs.uk/attributes/person_gender_code.html). */

- What was the sample size included in the final analyses? In the Abstract and at the beginning of the Results and Discussion sections, it is stated that it was 3,340 but, according to the information provided in the Methods and Table 1, I guess the correct number is 3,350.

/* Thank you for pointing out this error – the correct figure is indeed 3,350, and we have edited the Abstract, Results and Discussion to reflect this. */

- It would be useful to provide a little bit more thorough description of the NLP algorithm employed. I am not an expert in NLP, and I wonder how the algorithm deals with those cases in which a particular word appears in the middle of a negative statement (e.g., “the patient reported no cannabis consumption”) in the electronic record. I assume the algorithm is able to detect and do a correct classification of such instances but adding more information about how the algorithm works may help the readers. Alternatively, a link to the algorithm, if it is open-source, or a reference to a previous manuscript in which this algorithm is described in depth, could be useful as well.

/* Thank you for your feedback, which is shared by Reviewer 3. We have added the following sentences and references to the Methods section (Page 8): “CRIS offers a suite of NLP algorithms that are able to create structured data from unstructured free text through detection of key words and phrases. In brief, the algorithms are developed by applying cross-validated Support Vector Machines to a human-annotated training corpus of ‘positive’ (e.g. ‘UDS shows cocaine’), ‘negative’ (e.g. ‘does not use cocaine’) and irrelevant (e.g. ‘cocaine is a highly addictive drug’) examples for each symptom and substance. Their performance is then quantified against unseen data in an iterative process of development and testing.¹ The full library of NLP algorithms available within CRIS and details of their development are provided on the CRIS website.²” */

Reviewer: 2

Competing interests of Reviewer: None declared

Reviewer: 3

Dr. Carla Comacchio, University of Verona

Comments to the Author:

This is a study on the impact of gender and substance abuse on symptomatology at psychosis onset.

I have the following observations:

INTRODUCTION:

-NLP techniques are just mentioned, but a short paragraph on them could be useful

-objectives are not clearly stated and hypotheses are completely missing

/* In light of similar feedback from Reviewer 2 we have expanded on NLP techniques and their relevance to CRIS data in the Methods section, as we outline above. We have also added further information to the Introduction to highlight how these techniques differ from simple search queries (Pages 5-6): "These techniques use machine learning to learn from human-annotated examples and are able to navigate term negation (e.g., "No thought disorder elicited") and mentions irrelevant to what is being measured (e.g. "His father has thought disorder")."

We have added our hypothesis to the Introduction (Page 5): "We hypothesized that there would be differences between the genders in terms of symptoms and substance use recorded, and that these might be impacted by age at onset, ethnicity and substance use." */

METHODS:

-specify here which substances are detected by your algorithms (they are specified somewhere else in the paper)

/* We have clarified which substances are detected by the algorithms in the Methods section "Extracted substance data concerned cocaine, cannabis, amphetamine and MDMA." */

DISCUSSION:

-please, expand the section on ethnicity and age of psychosis onset, especially if they are included in your objectives

/* We expand on the role of ethnicity and age of psychosis onset and ground them in the research context on Page 12 of the Discussion: "Strikingly, all significant gender differences (bar tangential speech) remained after controlling for age and ethnicity. This suggests that gender differences in symptoms at presentation are independent of age and ethnicity. To our best knowledge, no studies have accounted for ethnicity in analyses of gender symptom differences,^{5,11,12,34} despite previous research indicating a significant association between ethnicity and risk of psychosis.³⁵ The lack of effect of referral age on gender differences is consistent with other studies' findings.^{6,32,34}" */

-the section on gender differences in symptoms presentation that become insignificant after controlling for substance abuse is nice and could be discussed more extensively

/* We have added some detail to the Discussion to highlight the lack of research in this area (Page 13): "This suggests that clinicians should take the role of illicit substance use into account when identifying strategies for symptomatic relief. In general, there is little research available into the ways in which substance use affect FEP presentation at the symptom level." */

Reviewer: 3

Competing interests of Reviewer: None declared

1. Jackson RG, Patel R, Jayatilleke N, et al. Natural language processing to extract symptoms of severe mental illness from clinical text: the Clinical Record Interactive Search Comprehensive Data Extraction (CRIS-CODE) project. *BMJ Open*. 2017;7(1):e012012. doi:10.1136/bmjopen-2016-012012

2. CRIS Natural Language Processing Applications Library.
<https://www.maudsleybrc.nihr.ac.uk/facilities/clinical-record-interactive-search-cris/cris-natural-language-processing/>. Accessed July 23, 2020.